# How RNases Shape Mitochondrial Transcriptomes

**DOI:** 10.3390/ijms23116141

**Published:** 2022-05-30

**Authors:** Jérémy Cartalas, Léna Coudray, Anthony Gobert

**Affiliations:** Institut de Biologie Moléculaire des Plantes, CNRS, Université de Strasbourg, 12 rue du Général Zimmer, 67084 Strasbourg, France; jeremy.cartalas@etu.unistra.fr (J.C.); lena.coudray@etu.unistra.fr (L.C.)

**Keywords:** mitochondrial gene expression, endoribonuclease, exoribonuclease, RNA degradation, RNA maturation

## Abstract

Mitochondria are the power houses of eukaryote cells. These endosymbiotic organelles of prokaryote origin are considered as semi-autonomous since they have retained a genome and fully functional gene expression mechanisms. These pathways are particularly interesting because they combine features inherited from the bacterial ancestor of mitochondria with characteristics that appeared during eukaryote evolution. RNA biology is thus particularly diverse in mitochondria. It involves an unexpectedly vast array of factors, some of which being universal to all mitochondria and others being specific from specific eukaryote clades. Among them, ribonucleases are particularly prominent. They play pivotal functions such as the maturation of transcript ends, RNA degradation and surveillance functions that are required to attain the pool of mature RNAs required to synthesize essential mitochondrial proteins such as respiratory chain proteins. Beyond these functions, mitochondrial ribonucleases are also involved in the maintenance and replication of mitochondrial DNA, and even possibly in the biogenesis of mitochondrial ribosomes. The diversity of mitochondrial RNases is reviewed here, showing for instance how in some cases a bacterial-type enzyme was kept in some eukaryotes, while in other clades, eukaryote specific enzymes were recruited for the same function.

## 1. Introduction

Mitochondria are organelles that are a hallmark of the eukaryotic cells, where they support the energy supply through respiration. The production of energy like ATP (adenosine tri-phosphate) is done thanks to the electron transfer chain located in the inner membrane of the mitochondria. The mitochondrial respiratory complexes are in most cases encoded by both mitochondrial and nuclear genomes. Indeed, mitochondria possess a genome that can come in different sizes and organizations across the eukaryotic phyla [1]. This genome codes for proteins of the OXPHOS (Oxidative Phosphorylation System) complexes, and can also code for parts of other complexes or pathways like the translation machinery (rRNAs, tRNAs). Usually, except maturases and sometimes the RNase P RNA component, no ribonucleases are encoded in the mitochondrial genome. The post-transcriptional and degradation events involve only imported RNases that are encoded in the nucleus.

Transcription of the mitochondrial genomes varies between groups of organisms [2,3,4,5,6]. Of course, this transcription might depend on the structure of the genome, but it can also depend on the number of promoters present on the genome. In animals, like humans, the mitochondrial genome master circle is transcribed as long polycistronic transcripts from both strands (light and heavy). In fungi, a more basal branch in Opisthokonta, two or more polycistronic transcripts are produced, the least number being the ancestral state. In plants, like *Arabidopsis thaliana*, the mitochondrial genome is transcribed from multiple promoters producing numerous transcripts that are mono- or polycistronic. In the sister clade of plants, Chlorophyta, in the most studied organism, *Chlamydomonas reinhardtii*, two polycistronic transcripts are produced. In trypanosomatids, the mitochondrial genome is composed of maxicircles and minicircles. Polycistronic transcripts are produced from both strands of the maxicircles, while the minicircles that are coding guide RNAs are monocistrons.

All in all, the diversity of transcripts produced needs maturation by diverse sets of ribonucleases. While some RNases are common across the different phyla of eukaryotes for maturation or degradation of RNAs, some of them are only necessary in specific phyla. Maturation and degradation of RNA can be mediated by endoribonucleases that catalyze the cleavage of single (ssRNAs) or double stranded RNAs (dsRNAs) at an internal phosphodiester bond of the RNAs, or by exonucleases that trim ssRNAs from the 5′ end (5′ to 3′ exoRNases) or from the 3′ end (3′ to 5′ exoRNases). The orientation of the degradation can be important to avoid the production of spurious short RNAs. RNases are found in different forms: single proteins (such as RNase III, RNase A and protein-only RNase P), homomeric or heteromeric protein complexes (PNPase, mitochondrial RNase P in animals), or functional ribonucleoprotein complexes (such as RISC complex and the ancestral type of RNase P). A catalytic function can also be mediated by different forms of RNase (RNase P examples cited above). RNases can also interact with protein partners in order to mediate recognition of the substrate or increase their catalytic specificity.

This review is describing the knowledge on the different endoribonucleases and exoribonucleases known to date that are implicated in the maturation and degradation of mitochondrial RNAs and how their combined work shapes the mitochondrial transcriptomes across eukaryotes (Figure 1). A full view of the diversity of these mitochondrial RNases across the eukaryote diversity is presented in Table 1.

## 2. Mitochondrial RNases Involved in RNA Maturation

### 2.1. Diversity of RNase P Enzymes

The majority of Opithokonthes organisms use the punctuation model for the maturation of the polycistronic primary transcripts where maturation of tRNA extremities liberate coding and non-coding RNAs [31]. In addition, even if other eukaryote clades do not use this model, tRNAs or tRNA-like sequences (TLS) can account for the maturation of some polycistrons or mRNA extremities [32,33]. Hence, the two endoribonucleases that are responsible for the maturation of tRNA precursors are of major importance. The first one, RNase P (Ribonuclease P) are endoribonucleases involved in cleaving the phosphodiester bond at the 5′ end of tRNA precursor mainly between position −1 and +1, and in some cases between −2 and −1. RNase P is an ubiquitous enzyme (one exception known), but groups different unrelated enzymes holding similar functions [7]. In eukaryotes, RNase P are localized in the nucleus, mitochondria and plastids in order to perform the maturation of tRNAs and TLS in these compartments where tDNA genes and TLS-containing genes are transcribed.

As mentioned earlier, RNase P is not a homogeneous group of enzymes. The first discovered RNase P is a ribonucleoprotein (RNP) complex containing a ribozyme in *Escherichia coli* [34], and RNP RNases P were subsequently found in eukaryotes where the number of proteins associated to the ribozyme increased from one to ten. Mitochondria are remnant of alpha-proteobacteria engulfed in a proto-eukaryotic cell, so it is not surprising that some mitochondrial genomes have retained the gene coding for the RNA moiety of the RNP RNase P [7]. This is true for several fungal lineages, basal Chlorophyta algae and jakobids [7]. The protein moiety is often encoded in the nucleus and imported from the cytosol [35,36,37]. The RNA moieties in these organisms can be very similar to the bacterial counterpart, but still to date no functional proof of activity was obtained in vitro with solely these RNAs supplemented or not with their proper protein subunits. Two possibilities are offered: either these are not functional enzymes, or more protein partners are needed and yet to be discovered. The RNP RNase P in mitochondria seems to be replaced over evolution by another type of RNase P with a single protein component: PRORP. Indeed, even if discovered as a protein complex in human mitochondria [38], there is no doubt that in the remaining phyla PRORP is active without any helpers [32,36,39,40]. PRORP has “invaded” various phyla and is present in metazoan mitochondria solely, streptophytes and most of the Chlorophyta algae mitochondria, the TSAR group contains mainly organisms with PRORP enzymes that are predicted to be localized in mitochondria, and finally the euglenozoan mitochondria (*Trypanosoma*) [7]. The maturation of tRNA precursors by RNase P in mitochondria, whose ancestor were bacteria with RNP RNase P, seems to have fallen under the control of the nucleus where PRORP is encoded. The process might be cost effective compared to a multipartite RNase P where the catalytic sub-unit is encoded in the mitochondria. The control of the RNase P expression can also be coordinated with the expression of other enzymes responsible for the maturation of tRNA that are also encoded in the nuclear genome. This latest fact could be true if a single PRORP was directed only to mitochondria, but in plants like *A. thaliana*, PRORP1 is dual targeted to mitochondria and plastids [32], and in the chlorophyte *C. reinhardtii*, a single gene codes for the three locations (nucleus, mitochondria and plastid) [40].

The different types of RNase P are divided in two main functional domains, which are the specificity domain and the catalytic domain. Of course, one being a ribozyme and the other a protein, they are of different nature. In the case of the RNP RNase P, several tridimensional (3D) structures were obtained. The simple dual-components 3D structure of the RNP RNase P from the bacteria *Thermotoga maritima* was resolved in 2010 [41]. The multicomponent (RNA + 5 proteins) 3D structure of the RNP RNase P from the archeon *Methanocaldococcus jannaschii* was resolved in 2019 [42]. Finally, the more complex multicomponent RNP RNase P 3D structures from the nuclear yeast and human RNP RNase P were resolved in 2018 [43,44]. To date, no 3D structures of RNP RNase P from mitochondrial origin were resolved. Still, in most cases, we can imagine a bacterial-like enzyme with probably addition of one or more protein partners. Indeed, as explained earlier, no RNase P activity could be obtained with only the two classical components for mitochondrial RNase P [36,37].

In the case of the PRORP enzymes, the first domain (at the N terminus) is an RNA interacting domain made of pentatricopeptide repeats (PPR). This domain serves also as a ruler to position the catalytic domain at the right place. The second domain is a metalloribonuclease domain from the NYN (Nedd4-BP1 YacP Nuclease) family. Both domains are connected by a zinc binding domain which might be responsible to add flexibility to the structure [45,46]. The 3D structure of the *A. thaliana* mitochondrial PRORP1 was achieved in 2012 [47]. This structure is very similar to the *A. thaliana* nuclear PRORP2 [45,48] or human mitochondrial PRORP [49], even if this last one does not cleave tRNA precursors alone in physiological conditions. Several studies point out that only motifs 2 and 3 of the PPR domain are responsible for the interaction [45,50], but the structural results of Teramoto et al. (2020) with only the PPR domain of PRORP1 in association with a tRNA show that motifs 1 and 4 are also implicated in the interaction. The PPR1 motif was previously shown to possibly contact the substrate in experiment of substrate masking lysine residues of PRORP1 [51]. These repeats are binding G18-C56 nucleotides of the tRNA precursors as it was earlier suspected [52,53]. From the structural and catalytic activity studies, the mechanism of cleavage of PRORP was deciphered. The catalytic site contains two magnesium ions to promote the hydrophilic attack on the phosphodiester bond [47,54]. All in all, an eye-catching mimicry appears between the binding mechanism of the tRNA elbow and the catalytic mechanism of 5′ leader cleavage between the RNP RNase P and PRORP, making them a perfect example of convergent evolution of completely different molecules. The complete human mitochondrial RNase P is formed by three proteins: methyltransferase TRMT10C (MRPP1), short-chain dehydrogenase/reductase-family member HSD17B10 (MRPP2), and MRPP3/PRORP, that hold the catalytic activity [38]. The 3D structure of this protein complex was recently resolved with its substrate [46]. These authors show that HsPRORP interaction with tRNA precursor is not so intricate as other PRORP, which could help to accommodate the tRNA that have short D/T loops present in animal mitochondria. MRPP1 and MRPP2 would help at positioning the HsPRORP enzyme on its substrates.

Numerous kinetic studies on *A. thaliana* PRORP enzymes have been performed, and they have shown that the 5′ leader and 3′ trailer does not contain determinant for recognition and cleavage. However, the enzyme cleavage site can be modified if N-1 interact with N+73 [55,56,57]. The D/T loops and the acceptor stem are necessary for an optimal cleavage of PRORPs, while the anticodon stem-loop in completely dispensable [52,55,56].

The mitochondrial RNase P activity is essential in most, if not all, organisms. Knock-out mutants are not viable [32,58]. Other mutations on the RNase P components can lead to the generation of aberrant processing of tRNAs and mRNAs, and lead to various diseases. For example, in *Drosophila*, knockdown of mtRNase P components leads to a lower level of ATP production [58]. In humans, a defective mtRNase P appears to be the cause of various disorders and diseases including HSD10 [59,60].

### 2.2. Diversity of RNase Z Proteins

RNase P through the maturation of tRNA and TLS in the mitochondria is therefore essential, but to free the tRNA from its polycistronic transcript, or to simply mature a tRNA precursor with its own promoter, an extra enzyme is needed that is RNase Z. This enzyme is processing the 3′ end of the tRNA. The enzyme performs an endoribonucleolytic cleavage after the discriminator base (N73) [61]. CCA is usually not coded in the mitochondrial genome and added thereafter by a CCA tRNA nucleotidyltransferase. The absence of RNase Z leads to the accumulation of tRNAs with a 3′ extension in the mitochondria [8]. Contrary to RNase P, all known RNase Z activities seem to be mediated by the same enzyme family of metallo-beta-lactamases.

RNase Z are defined as enzymes belonging to the metallo-beta-lactamase family, removing the tRNA 3′ trailer by phosphodiester hydrolysis, and leaving a 3′-OH at the end of the matured tRNA and a 5′-P at the end of the cleaved trailer. However, two different forms of RNase Z were discovered. The first one called RNase ZS is present in archaea and most eukaryotes and bacteria (Elac1 in human). The second one called RNase ZL is only present in eukaryotes (Elac2 in human). Both forms differ in length, the longer form (L: 750–930 amino acids) being twice the size of the shorter form (S) and probably resulting from a duplication of the latter one. Indeed, the fact that both N-ter and C-ter parts of RNase ZL show sequence similarity to the RNase ZS (higher for the N-ter region) and that all RNases ZS enzymes analyzed were homodimers in solution, suggest that RNase ZL enzymes might have evolved from RNase ZS gene duplication to behave like a covalently linked dimer [62]. The overall data in the literature suggests that RNase ZL is responsible for the maturation of tRNA, while RNase ZS is responsible for tRNA recycling [63], but this theory cannot be exclusive because plastids contain only a shorter form of RNase Z for tRNA precursors maturation [9]. Until now in animals, the mitochondrial RNases Z identified are RNases ZL that are produced from the same nuclear gene as nuclear RNase Z thanks to an alternative translation initiation [10]. In earlier diverged Opisthokonta, like the fungus *Schizosaccharomyces pombe*, two genes code for RNases ZL with one coding for a mitochondrial isoform and the other coding for a nuclear isoform [11]. In plants like *A. thaliana* or rice, the genes are also duplicated. The two RNases ZL are both localized in mitochondria, with one of them being also localized in the nucleus [9]. In conclusion, mitochondria tRNA or TLS precursors 3′ end are to date only been found to be cleaved by the long form of RNase Z. The reason of this selection over the short form (contrary to the plastid) is still unknown.

The RNase ZS enzymes have been extensively studied showing the domains and motifs important for catalysis and tRNA binding [64,65]. In the case of RNase ZL that are present in mitochondria, structural studies of the yeast dual localized TRZ1 (mitochondria and nucleus) align with the biochemical studies [62,66]. The 3D structure of TRZ1 confirms that RNase ZL are probably a duplication-fusion of RNase ZS. In addition, the authors show that the N-ter divergent domain fold similarly as the C-ter domain in a β-lactamase structure. While the N-ter domain is inactive, it contains the necessary extension for tRNA binding. On the contrary, the C-ter domain is active but has lost the extension that can interact with the elbow of a tRNA precursor. The C-terminal domain contains the hallmarks of active RNase Z: the Zinc coordination motif HxHxDH, the PxKxRN and AxDx motifs, as well as the HEAT-α17-HST motifs that are crucial for tRNA binding and cleavage [62]. In a second paper, Ma et al. (2017b) show that TRZ1 interacts with another nuclease Nuc1 and a maturotase to form a stable heterohexamer. Since the mitochondrial Nuc1 is transferred to the nucleus upon apoptosis, TRZ1 is dual localized in the mitochondria and nucleus, and mutarotase is localized in the cytosol and nucleus, this complex could be localized in the nucleus. However, their results show that Nuc1 is inhibited upon binding to its partners. The authors hypothesized that two subcomplexes exist: one in the mitochondria (TRZ1-NUC1), and one in the nucleus (Mutarotase – Nuc1). These data show that TRZ1 has also protein interaction domains that remain to be determined precisely. Additional protein interactions could occur as the yeast mitochondrial RNase Z seems to be present in a supercomplex with RNase P and the RNA degradosome [67].

The mitochondrial RNase ZL contains only one catalytic site in the C-ter domain. The alignment of the 3D structure of TRZ1 with the homodimer of *Bacillus subtilis* RNase ZS with its substrates shows that while RNase ZS homodimer can accommodate the cleavage of two tRNA, the single TRZ1 can only accommodate one [62]. Hence, a single RNase ZL cleaves a single tRNA or TLS precursor. TLS cleavage by mtRNase Z was indeed shown to occur in *A. thaliana*, where several mitochondrial mRNA precursors contain TLS [9]. Therefore, mtRNase ZL are equally important for the maturation of mRNAs in various organisms whether the punctuate model apply or not.

In *Saccharomyces cerevisiae*, the mtRNase Z encoded by TRZ1 is essential, and a decrease in its expression leads to severe mitochondrial deficiencies. In *S. pombe*, both genes encoding RNases ZL are essential, but only the overexpression of the mitochondrial isoform (SpTrz2p) causes a characteristic abnormal phenotype [11,68]. In Drosophila, a first molecular study had shown that dRNase Z, the gene coding for both a nuclear and a mitochondrial RNase Z, was an essential gene since its knockdown caused growth arrest and early larval lethality [69]. A specific KO of the mtRNase Z allowed observation of impaired mitochondrial polycistronic transcript processing, increased reactive oxygen species (ROS), and a switch to aerobic glycolysis compensating for cellular ATP, leading to a disrupted cell proliferation without affecting viability in vivo [70]. In humans, it has been shown that mutations in ELAC2 coding for mtRNase Z were the cause of patients presenting with mitochondrial respiratory chain deficiency, hypertrophic cardiomyopathy (HCM) and lactic acidosis, and even prostate cancers [71,72].

### 2.3. MNU Proteins Make a Novel Category of Mitochondrial RNA Maturation Factors

In the maturation process of mitochondrial RNA, extra enzymes might be needed to trim the extremities at both 3′ and 5′ ends. Putative endonucleases might be implicated in the cleavage of 5′ extensions of mRNA in seed plants. Indeed, while the 3′ ends of mRNAs are fixed, the 5′ ends can be multiples due to the presence of different promoters producing different pre-mRNAs, and for each, different possible maturations. MNU1 and MNU2 were shown to impact the 5′ extremities in this case [73], and these enzymes could also be implicated in tRNA maturation [13]. These enzymes are composed of a NYN catalytic domain [74], two Lotus (OST-HTH) domains [75], an OHA domain (OST-HTH associated), and a putative WW domain. Until now, the catalytic activity of the NYN domain was not clearly demonstrated and there is no proof of an exo- or endonuclease activity. However, mutants show elongated 5′ extremity for some mRNAs [73]. MNU2 was also found interacting with the PRORP1 (RNase P) through its potential WW domain, and might be implicated in the maturation of tRNA precursors or TLS 5′ ends [13]. This type of enzyme has probably a common ancestor with the cytosolic MARF1 orthologue called DNE1 in Arabidopsis as they share the same type of catalytic domain and Lotus domains [76]. In addition, a lotus domain of the DNE1 enzyme was shown to interact with G quadruplexes in RNA [77]. However, it remains to be proved that Lotus domains of MNU enzymes can also bind these structures in vitro and in vivo.

Except the molecular phenotype presented, plants lacking MNUs do not show any macroscopic phenotype in standard growth conditions. This means that their absence has no real impact on the stability of the RNA and does not highly impact the level of mitochondrial translation or the ratio between long 5′ leader and short 5′ leader is low enough to not impact the translation.

### 2.4. KREX and KREN Ribonucleases Are Required for RNA Editing in Kinetoplastid Mitochondria

In kinetoplastids, the mitochondrial genome is peculiar and needs to be heavily edited. This maturation process is mediated by editosomes that contain exonucleases and endonucleases. We decided to describe both types of enzymes subsequently. Editing of mRNA precursors consists of insertion or deletion of uridines. For this purpose, a guide RNA or gRNA recognizes the sequence to be edited and a set of proteins forming an editosome performs the insertion or deletion. Two 3′-5′ exoribonucleases named KREX1 and KREX2 are found in the editosome.

KREX1 and KREX2 are responsible for U removal in the editosomes. The catalytic site belongs to the endonuclease/exonuclease/phosphatase (EEP) family, containing notably RNase H. To date and to our knowledge, no 3D structures are available for KREX enzymes, and no RNA binding domain has been predicted. The binding might be mediated through protein interactions with RBPs.

Editosomes seem to be very dynamic complexes with common components but also specific ones. Some authors have classified the editosomes in three complexes based on the type of endoribonucleases presents (KREN1, KREN2 or KREN3) that we will describe later on. KREX1 is exclusively associated with the endoribonuclease KREN1, while KREX2 is in the deletion subcomplex of all three editosomes [14,27]. In vivo studies of knock down genes were performed, and TbKREX1 and TbKREX2 underexpression have different outcomes. KREX1 knockdown results in the reduction of RNA editing, and consequently cell viability, while KREX2 knockdown does not affect editing and cell growth. However, the downregulation of both KREX1 and KREX2 leads to a worse growth phenotype compared to the single down regulations [14]. In *Leishmania major*, the catalytic site of KREX2 is not functional, supporting the statement that KREX1 is the major 3′ U exoribonuclease in editosomes.

As stated before, editosomes of kinetoplastids are composed of 3′ exoribonucleases and endonucleases. The endoribonucleases KREN1, KREN2 and KREN3 of *Trypanosoma brucei* are mutually exclusive per editosome, leading to classification of at least three different editosomes. These enzymes are responsible of the cleavage of the mRNA precursors pinpointed by the guide RNA at the 3′ of U to be edited.

KRENs contain a functional RNase III domain, a U1-like zinc-finger domain and a PUF domain [78,79]. The RNA recognition motifs can bind double stranded RNAs at the level of the pre-mRNA/gRNA.

The KREN enzymes seem to form heterodimers with non-catalytic orthologues like KREPB4 to KREPB10 [79,80,81]. This heterodimer would bind dsRNAs and cleaves only the pre-mRNA (one sub-unit functional). However, while a single KREN is present in an editosome, several proteins of non-functional catalytic domain are present. The diversity could give specificity to some substrates or edited loci. U1-like zinc finger domain is essential for KREPB4 but not for KREPB5 function. On the contrary, PUF motif is necessary for KREPB5 but not for KREPB4 function [79,82]. The presence of a RNase III domain is essential in the non-functional enzymes. The cleavage of the pre-mRNA takes place on an unpaired nucleotide immediately upstream of the gRNA-mRNA anchor duplex, leaving the phosphate on the 3′ end of the 5′ cleavage product (or the 3′ end of the gRNA), and enabling the U addition or removal at this location [83,84]. The cleaved strand required a minimal 5′ overhang of 12 nucleotides and a ~15 bp duplex with gRNA to direct the cleavage site [85].

Kinetoplastida, like *Leishmania* or *Trypanosoma* are parasites that target animals and humans. Targeting enzymes specific to this organism could lead to treatment of the disease. The mitochondrial editosome is quite unique to these organisms and understanding its function is therefore really important.

### 2.5. RAP Domain Nucleases Make a Diverse Family of Putative Mitochondrial RNA Maturation Enzymes

A group of RNA binding proteins with a putative ribonucleolytic site was defined in 2004 by Lee and Hong with a lot of representatives in Apicomplexans. They called this domain RAP for “RNA-binding domain abundant in Apicomplexans” [86]. Later bioinformatic analysis would show that this family is overrepresented not only in apicomplexan, but also in Dinoflagellates, and to a lesser extent in Chlorophytes [19]. These proteins are mainly predicted to be localized in organelles like mitochondria and chloroplasts. The proteins usually contain a RAP domain at their C-terminus, and the majority of them have helical repeats at their N-terminus. These helical repeats can be TPR, PPR, HPR or OPR (for tetra-, penta-, hepta- or octo-tricopeptide repeat). These domains are known for protein interaction (TPR) or RNA interaction (PPR, HPR, OPR). Until now, no 3D structure is available for the RAP domain, but models show that they adopt a PD-(D/E)-XK nuclease superfamily fold, and some RAP domains are best modeled on bacterial VSR endonuclease [87]. A conserved aspartate of the catalytic site in VSR and RAP was mutated into an alanine in order to get insight on the possible endonucleolytic activity of the RAP domain of FASTKD4, a human rap domain protein [87]. The expression of this mutated protein in the KO line did not rescue the molecular phenotype, showing that the RAP domain is possibly a nuclease domain. In humans, all RAP domain containing proteins are localized in the mitochondria and affect the mitochondrial transcriptome to a certain level [87,88,89]. These proteins have been implicated in the maturation of polycistronic mRNA junctions when no tRNA is present to be matured by RNase P and RNase Z and non-coding RNA degradation [90]. In addition, FASTKD2 was shown to target 16S rRNA [91]. This is not the sole RAP domain protein to target ribosomal RNA, as in plant chloroplasts where the sole RAP domain protein is implicated in 16S ribosomal RNA maturation [92], and in *Plasmodium*, PfRAP01 and PfRAP21 knock-down proteins affect mitoribosomes through their interaction with rRNA fragments [93]. These last authors advance the hypothesis that the multiplication of RAP proteins in Apicomplexa and Chlorophytes is linked to the highly fragmented rRNA encoded in their genomes. In conclusion, more work is needed to prove that the RAP domain has a catalytic function, but it remains evident that the domain is necessary for the function of the protein in modulating mitochondrial RNAs.

Mutations of FASTK enzymes were associated to adult onset MELAS (mitochondrial encephalomyopathy, lactic acidosis and stroke-like episodes)-like syndrome [94]. FASTK genes were found upregulated in different types of cancer, and among the FASTK family, FASTKD3 mutations were found in 33 cancer types [95].

## 3. Mitochondrial RNases Involved in RNA Quality Control and Degradation

### 3.1. PNPase Plays a Central Role in Mitochondrial RNA Degradation

Mitochondrial RNA seems to be prone to maturation and degradation occurring mainly from the 3′ ends. The main enzyme for this exonuclease activity is the polynucleotide phosphorylase (PNPase) in some protists, animals and plants.

Animals and plants differ in their maturation process for mRNAs. Indeed, 3′UTR are present in most plant mRNAs because no tRNA or TLS are there to specify their ends. The 3′ end can therefore be trimmed by 3′-5′ exoribonucleases. Even if polycistronic transcripts are produced with tRNAs, the 3′ end of mRNAs are not specifically close to these tRNAs, and the intergenic sequences need to be degraded. In consequence, plant mitochondrial PNPase (mtPNPase) is not only involved in degradation, but also in maturation of the transcripts. The mtPNPases are very similar to their bacterial counterparts. They are 3′-5′ phosphorolytic exoribonucleases.

Metazoan PNPase monomers are very similar to bacterial ones and contain two RNase PH-like domains, followed by a KH domain and an S1 domain [96]. In seed plants, the mtPNPase monomers are longer and contain an additional S1 domain at their N-termini, as determined by bioinformatics. The enzyme is constituted from three monomers assembled to form a ring shape enzyme holding a central channel allowing the RNA to pass through and contact the internal catalytic site. The core of the PNPase is formed by its catalytic domain with the KH and S1 domains protruding like three tails. These tails are mobile and this mobility seems to be essential to catch the RNA substrates in order to direct them in the phosphorolytic channel, as shown by the cryo-EM structure of the bacterial PNPase [97]. This mobility is probably also present in mtPNPases KH-S1 tails, and it would be interesting to understand why seed plants need longer RNA interacting tail length (KH-S1-S1).

As stated before, mtPNPases are mainly RNA degrading enzymes. MtPNPase targets various types of mitochondrial single stranded RNAs: coding RNAs [23,98], non-coding RNAs [99,100], endonucleases by-products [101], and antisens RNAs [101,102] in different eukaryotes. The enzyme is also implicated in double stranded RNAs degradation [103,104]. Per se, PNPase can only degrade ssRNAs, and its association with the SUV3 helicase to form the mtEXO complexe make it possible to linearise the ssRNA structures. Indeed, in the mitochondrial matrix, human mtPNPase forms a 330 kDa heteropentameric complex with the human SUV3 helicase with a 2:3 molar ratio of SUV3/mtPNPase [105,106]. This complex functions like the RNA degradosome found in bacteria. The PNPase enzymes share their structures and their phosphorolytic activities with the Archaea and plant cytosolic and nuclear exosomes [107,108]. This activity is mediated by the active site in the hexameric RNase PH ring, in which an inorganic phosphate attacks the phosphodiester bonds, releasing 5′ di-phosphate ribonucleotides. The remaining products are usually around 4 N long depending on the species [109].

MtPNPases prefer specific sequences to bind at the 3′ end of RNAs. The human mtPNPase binds preferentially to poly(U) and poly(G). However, several studies show that poly(A) 3′ ends are also targeted in human mitochondria like in bacteria (i.e., Toompuu et al. (2018)). In plants, poly-adenylation is a degradation signal and the plant mtPNPase is implicated in this degradation [23]. The conserved GxxG motif of KH domains seems to be essential for RNA binding, while the S1 domain could be dispensable for substrate binding [109]. However, mutation in the S1 domain leads to severe pathologies [110], meaning that it is not completely dispensable in vivo. This domain at least in mammals could be implicated in the RNA import in mitochondria [111].

In humans, several functions were found for mtPNPase. Indeed, it was found in both the IMS and the matrix of mitochondria [112]. Beside its RNase activity, mtPNPase is also implicated in mitochondrial genome maintenance in the matrix [113]. A part of mtPNPase is located in the IMS, where it is responsible for the import into the mitochondria of nuclear produced RNAs. In addition, a recent study shows that mtPNPase, and especially its S1 domain, has an undefined role in the decrease of the oxidation of mitochondrial RNAs [114]. Other study found mtPNPase in the cytosol interacting with an oncoprotein [115]. Hence, this enzyme, if mutated could have pleotropic effects. Accumulation of RNA intermediates and an increase of RNA half-life is observed when the gene is silenced [112]. Mouse mtPNPase loss leads also to mitochondrial genome loss [116], which could be principally due to harmful R-loop formation [113]. In plants, the loss of mtPNPase is lethal.

Rius et al. (2019) have compiled all the published and new cases of human mtPNPase mutations (24 patients). The symptoms comprise muscle tone abnormalities, feeding difficulties, various severity of hearing loss principally and occasionally regression, choreoathetosis, visual loss and cataracts [110]. Several studies showed that the mtPNPase trimer is not formed in the context of these mutations. Mutations in the RNase PH domains, therefore impacting RNA binding and catalysis, seems to be the more severe. Some mutations impact the trimerization of mtPNPase, leading to the formation of a dimeric form with probably reduced RNA import and degradation [111]. Indeed, in this structure configuration, the GxxG motif of the KH domain is less accessible for RNA binding, making it more difficult to target the RNA in the processing channel. In addition, the S1 domain probably involved in stem-loop binding of imported substrates is positioned along the RNase PH domains, impairing RNA import [111].

### 3.2. DSS1 Proteins Are Specific Mitochondrial RNA Degradation Enzymes in Fungi and Kinetoplastids

Contrary to metazoan and plants, yeast and trypanosomatids do not possess mtPNPase. The degradation of mitochondrial RNAs in these organisms rely on DSS1 enzymes. This enzyme, like PNPase in other organisms, forms a complex called mtEXO (mitochondrial exosome) with a helicase.

DSS1 enzymes belong to the RNase II–RNase R family composed of 3′-5′ exoribonucleases that are processive enzymes. These are metal dependent enzymes. Razew et al. (2018) have resolved the 3D structure of the mtEXO complex of *Candida glabrata* formed by DSS1 and the RNA helicase SUV3, an NTP-dependent RNA helicase related to the DExH/D superfamily. DSS1 is composed of a β-barrel, a winged helix (WH) domain, a HTH domain, a RNB catalytic domain and a S1 domain. The β-barrel, HTH, WH, and S1 domains of DSS1 form a funnel opening into the channel formed by the RNB domain. The substrate is thought to move through the channel with the catalytic activity occurring at its basis. The 3D structure of mtEXO also revealed that DSS1 has an N-terminal β-barrel, WH and HTH that replace the typically observed cold-shock domains CSD1 and CSD2 found in RNase II [21]. The authors show a complex with a 1:1 ratio between DSS1 and SUV3.

DSS1 enzymes target RNAs with 3′ overhangs and produce 5′ monophosphate rN, leaving 4–6 Nt undigested products [21]. mtEXO action necessitate both enzymes DSS1 and SUV3. The loss of one of the components results in the loss of RNA degradation activity [117]. In yeast, most mRNAs terminate with a dodecamer sequence element. The PPR protein Rmd9 was found to bind this element and stabilize mRNAs [118]. This stabilization occurs by blocking the action of mtEXO at least in vitro. In yeast, another PPR protein Ccm1p seems on the contrary to mark the extremity of the 15S rRNA for cleavage when present. It might remodel the RNA structure to give access or recognition of the substrate by an unknown endonuclease, perhaps like it happens in plants where a PPR when binding modifies the local structure of the RNA in order to produce a TLS that is cleaved by PRORP [119]. The authors exclude DSS1 as the main 3′-5′ exoribonuclease responsible of the degradation of the cleaved extremity of the 15S rRNA, and hypothesize an action of Dis3p that could be double localized in the mitochondria [120]. The yeast transcripts are not tailed with poly(A) or poly(U) that could mark them for maturation or degradation by DSS1, but in *Trypanosoma*, the gRNAs produced from the minicircles are U-tailed and there is a functional coupling between the action of RET1 TUTase and DSS1 (TbDSS1 or KDSS1). Like in yeast, KDSS1 is not functional alone and needs to be integrated into the mitochondrial processosome (MPsome) [121,122], or associates with TbSUV3 in a degradosome [22]. The TbmtEXO plays a role in the degradation of intermediates and by-products of 12S rRNA maturation [123].

Although DSS1 is found in yeast and trypanosomatids mitochondria that are devoid of PNPase, a RNase R enzyme (RNR1) is also found in plants mitochondria, even if they possess a PNPase [23]. This enzyme is dual localized in mitochondria and chloroplasts, and has been extensively studied in the chloroplast at the functional level. The only function established for RNR1 is a role in the maturation of the 3′ extremity of atp9 mRNA. The enzyme is responsible for the final trimming of atp9 3′ end after the primary action of mtPNPase [23]. Hence, RNR1 seems to have a role in quality control of the 3′ ends, polishing the work of the PNPase.

The yeast DSS1 is a factor reducing the life span. The increased lifespan in the dss1 mutant seems to be coupled with the low amount of mtDNA and ROS. However, its overexpression does not reduce life span [124].

### 3.3. Participation of Oligoribonucleases in Mitochondrial RNA Degradation

As mentioned before, the 3′-5′ exoribonucleases are not degrading the whole RNA, partly due to the length of their internal channel (catalytic site at the basis of the channel and RNA binding domain feeding the channel at the top). They leave few oligoribonucleotides from around 4 to 12 in length. These oligoribonucleotides are then taken in charge by oligoribonucleases to be completely degraded. The oligoribonucleases seem to be present in most eukaryotes, but to our knowledge, only the ones in yeast and humans have been shown to be mitochondrial.

The best studied mitochondrial oligoribonuclease is the human REXO2 [24]. This enzyme belongs to the 3′-5′ exonuclease DEDDh superfamily. REXO2 forms a homodimer as observed in the 3D structure [125,126,127]. The authors show that the dimerization is crucial for activity using mutated residues implicated in this process [126,127]. The result is in agreement with the structure where both subunits contain residues forming an aromatic clamp that interact with the substrate [126,127].

The degradation of nanoRNAs by REXO2 in mitochondria prevents their accumulation that inhibits the RNA degradosome function. [127]. Therefore, a strong functional link exists between PNPase-SUV3 degradosome and REXO2. REXO2 eliminates substrates that could serve as primers to initiate transcription on promoters and other parts of the genome creating aberrant transcription [127]. The substrate preference of REXO2 is dinucleotides [126,127], like its bacterial ancestor orn [128]. However, longer substrates can be degraded [127]. REXO2 catalytic site is formed by the residues D47, E49, D147, D199 (DEDD). The four residues are coordinating two Mg^2+^, and their mutation impairs the function of the enzyme [126]. The conserved histidine of DEDDh enzymes, that is H194 in REXO2, seems to make a shift of 180° in order to position itself in the catalytic site upon binding of the substrate [126]. H194 would activate a water molecule for the nucleophilic attack on the phosphodiester bond [125]. RNA binding is mediated by π–π stacking interaction primarily with the last two 3′-end nucleotide bases with Leu53, Trp96, and Tyr164 of REXO2 [125].

REXO2 has been established as essential; indeed, its silencing resulted in stellate cell morphology, affected cell growth, loss of the mitochondrial network to punctata, and decreased mtDNAm as well as mtRNA levels and translation [129]; in addition, REXO2 knock-out, which was lethal to mouse embryos, altered gene expression patterns and increased the pool of available dinucleotides [126].

### 3.4. Other Types of Mitochondrial RNase Enzymes Involved in RNA Degradation

Other 3′-5′ exoribonucleases from the DEDD family are active in mitochondria but their occurrence is often restricted to one phylum of eukaryotes.

An RNase D-like enzyme was observed in mitochondria of trypanosomatids [25] and its structure was partly resolved [130]. This RNase D-like enzyme possesses the typical catalytic domain, but unconventionally has an CCHC zinc-finger domain at its N-terminus. This domain is composed of four successive zinc-finger motifs that are binding the target. Zimmer et al. (2011) observed that the downregulation of *Trypanosoma brucei* RNase D (TbRND) leads to an increase in the poly(U) tail lengths of guide RNAs, but not of rRNA. Upon processing, the enzyme leaves 2 to 3 Us next to the gRNA sequence. This feature is not understood to date. It might be due to the presence of RNA-binding proteins on the gRNA sequence or the very nature of the enzyme. Indeed, the catalytic domain accommodates three ribonucleotides in a pocket [130], while the two following ribonucleotides either do not interact or interact a lot with the RNase D domain. The next ribonucleotides are interacting with the two first zinc finger motifs. Truncating the zinc finger domain abolishes the activity of the enzyme, and leaving only one zinc finger motif strongly reduces its activity. With two zinc finger motifs, the activity is better but does not reach the one of the WT enzyme [130]. The zinc-finger domain is therefore necessary for the binding of the poly(U) tails in order to feed the catalytic domain.

An enzyme from the DEDDy family that trim the 3′ end of polyadenylated RNA is found in plant mitochondria. This enzyme is a deadenylase nuclease called PARN, that is usually not observed in this organelle in other eukaryotes like human. The function of this enzyme is conserved from basal streptophytes to angiosperms [28,131]. A recent review details the function of this enzyme [132], and only information about its possible structure linked to its function are presented here. From the structural studies of its human orthologue that is not localized in the mitochondria, we can extrapolate that the mitochondrial PARN works as a homodimer. Each sub-unit is composed of at least three functional domains: R3H, DEDD nuclease and RRM [29,133]. The R3H domain of the first sub-unit forms a lid above the catalytic site of the second sub-unit (and vice versa). This lid encloses the poly(A) in the RNA binding cavity. The RRM domain of HsPARN binds the poly(A) and the m^7^G cap. This RRM seems to be conserved in the plant PARN and might hence contact only poly(A) as there is no 5’ caps at the extremity of mitochondrial RNAs. The human PARN is a processive enzyme, and the overaccumulation of poly(A) RNAs in mitochondrial PARN downregulated mutant suggests that this is also the case in plants. PARN function is essential and its absence is lethal for the plant.

### 3.5. Functions of RNase T2 Enzymes in Mitochondria

In order to scavenge unwanted mitochondrial RNA, different types of RNases are used. RNase T2 enzymes are widespread in the different phyla of the tree of life and in viruses [134]. In humans, only one member is present and found in different localizations of the cell. Recent publications report this enzyme to be also localized in the IMS of mitochondria [18].

RNase T2 is a member of the Rh/T2/S family of acidic hydrolases. These are transferase type RNases that catalyze the cleavage of single-stranded RNA through a 2′,3′–cyclic phosphate intermediate. RNA cleavage by RNase T2 is a two-steps mechanism: transphosphorylation and hydrolysis. The final products are mono- or oligonucleotides with a terminal 3′-phosphate [134]. T2 ribonucleases generally cleave at all four bases. The structure of RNASET2 in humans was published in 2012 [135]. RNASET2 has a typical fold of seven α-helices and eight β-strands that constitute an α + β motif. Four disulfide bridges consolidate the structure. Some potential Zn^2+^ binding sites were predicted. The enzyme is known to be inhibited by zinc and copper. The catalytic site of RNases T2 is formed by two motifs, CASI and CASII, whose His residues facilitate the reaction. RNA binding in the catalytic pocket is pretty well defined, but how RNAs are targeted by the enzyme is, to our knowledge, still unknown. In animals, latest reports show that mtPNPase seems to be targeted to the IMS, taking in charge the RNA for transport across the inner membrane [136]. Based on these previous results, Liu et al. (2017) proposed a mechanism for mtRNA decay by RNase T2 that is also partially localized in the IMS [18].

Patients with mutated RNase T2 gene show white matter disease of the brain appearing in early infancy. Their diagnosed conditions are cystic leukoencephalopathy, as they show psychomotor retardation, spasticity and epilepsy [137]. However, it is unknown if the conditions are linked to the particular localization of RNase T2 in the IMS.

### 3.6. Mitochondrial RNases That Might Be Involved in RNA Quality Control Pathways

In human mitochondria, a deadenylase from the endonuclease/exonuclease/phosphatase (EEP) family (PFAM: PF03372) is found. This enzyme is a 2′-phosphodiesterase called PDE12. PDE12 is composed of two main domains: an N-terminal domain forming a two-layer β-sandwich, immunoglobulin (Ig)-like that might function in RNA binding or RNA structure unfolding [138,139], and a C-terminal domain typical of the EEP catalytic domain. In the EEP domain, a single Mg^2+^ ion was found in the 3D structure [139]. Using PDE12 overexpressors, it was first thought to remove mitochondrial mRNA poly(A) tails [140]. However, the use of a pde12 mutant showed that the enzyme has only a subset of RNA targets in the mitochondria. Indeed, PDE12 only seems to control spurious adenylation of the 3′ end of 16S mt-rRNA and mt-tRNAs [26]. PDE12 main target is therefore non-coding RNAs and not mRNAs in the mitochondria, and acts on the quality control of their 3′ ends. The enzyme seems to have a slight preference for RNA poly(A) and poly(U) over poly(C), and poly(G) is a very poor substrate. PDE12 does not digest DNA substrate and needs the 3′ hydroxyl group on RNA for activity [138,140]. The enzyme can also degrade 2′–5′ oligoadenylate substrate [138]. In the overexpressed line of PDE12, translation of all mitochondrially encoded respiratory chain subunits was compromised, while in the mutant line *pde12*, spurious polyadenylation of 16S rRNA did not affect translation, but spurious polyadenylation of some tRNAs would reduce their aminoacylation and induce ribosome stalling at the corresponding codon [26,140]. Of note, PDE12 was also reported to degrade 2′-5′-oligoadenylate, an unusual oligonucleotide that activates RNase L to increase viral resistance [138]. Another enzyme from the same EEP family, Nocturnin, was also shown to be partially targeted to the mitochondrial matrix depending on the cell lines [141,142,143]. Nocturnin seems to localize in mitochondria in muscles (skeletal and heart) and partly in the brain [141]. In Nocturnin KO in the liver, longer poly(A) tail cytosolic transcripts are related to mitochondrial function [144]. This enzyme was first shown to process mRNAs as PDE12 [145,146], but recent reports suggest that this enzyme does not have exoribonuclease activity against poly(A) in vitro [147,148], but could dephosphorylate NADP(H) to NAD(H) [142]. Therefore, Nocturnin is probably an NADP(H) 2′-phosphatase and not a deadenylase.

Another 3′-5′ exonuclease that we would like to present is Myg1 [30]. This enzyme is dual localized in the nucleus and mitochondria. In detail, the results of these authors show that a pool of Myg1 in the cytosol might be post-translationally regulated to be targeted to both mitochondria and the nucleus. Myg1 is found in fungi, humans and plants. In the mitochondria, the enzyme targets the 3′ termini of mRNAs and rRNAs, while in the nucleus, the transcripts coding for mitochondrial targeted proteins are also a target of this enzyme. This enzyme could participate in the coordination of the function between mitochondria and nucleus.

## 4. Miscellaneous Functions of Mitochondrial Ribonucleases

### 4.1. RNase H Enzymes Are Involved in Mitochondrial DNA Maintenance

Ribonucleases, in addition to mature or degrade transcripts, can also participate in the genome maintenance and replication. Indeed, mitogenome replication makes use of small RNA primers that need to be degraded upon use. RNase H is an enzyme capable of cleaving RNAs only in an RNA/DNA hybrid duplex by releasing 3′-OH and 5′-P ends. This enzyme is present in all three kingdoms of life and can be targeted in different organelles.

RNase H belong to the endonuclease/exonuclease/phosphatase (EEP) family. RNase H activity has been detected in the mitochondria of several organisms. Most papers describing this activity refer to RNase H1, while a very low proportion suggests the presence of RNase H2 in the mitochondria, particularly in *Leishmania* [149].

In a very simplified way, RNases H1 are generally composed of an MTS, a DNA/RNA hybrid binding domain (HBD), a connection domain (CD), and an RNase H domain. Although the N-ter and C-ter regions appear to be highly conserved, variation in the size of these domains can be observed (particularly at the level of the CD), as well as the complete absence of MTS (as in *C. elegans* and *S. cerevisiae*) [150]. In many organisms, including humans, mice, Drosophila, and the trypanosomatid species *Crithidia fasciculata*, a single gene encodes both nuclear RNase H1 and its mitochondrial isoform [150,151,152]. On the contrary, in other organisms such as *A. thaliana*, RNase H1 is encoded by three different genes and then localized in the nucleus, mitochondria or the chloroplast [153].

MtRNases H1 are versatile and have multiple roles in mitochondrial DNA metabolism [154]. Firstly, it appears to be essential for mtDNA maintenance since a KO of RNase H1 in mice causes embryonic lethality around 10 dpc and mtDNA depletion [150]. Secondly, it seems to be necessary for the proper replication of mtDNA, since the absence of RNase H1 in embryonic fibroblasts led to the retention of primers, particularly one at the prominent lagging-strand initiation site (Ori-L). Those primers then represent an obstacle for the mtDNA Pol γ and lead to the generation of a double-stranded break at the origin when the resulting gapped molecules are copied. These results suggest that the necessity of Rnase H for mtDNA replication is due to its ability to remove RNA primers at the origin of replication [155,156]. Thirdly, it is involved in the processing of R-loops (3-stranded nucleic acid structures with a DNA/RNA hybrid strand and a single DNA strand) in different organisms, even in *S. cerevisiae*, where RNase H1 was not described as mitochondrial [157]. This processing will promote various processes, notably the maintenance and replication of mtDNA (as mentioned earlier); a model where R-loop processing by RNase H1 directs origin-specific initiation of DNA replication has even been proposed in humans [158,159,160].

Although RNase H1 is not essential in prokaryotes and lower eukaryotes, it is essential for the survival of higher eukaryotes since its alteration or absence is the cause of disease or embryonic death of the individual [152,153,161,162].

### 4.2. Putative Mitochondrial Functions of RNase MRP

RNase MRP (mitochondrial RNA processing) is a ribonucleoprotein complex found in most, if not all, Eukarya whose components are nuclear encoded. It is composed of a catalytic RNA (RMRP in human) and around ten protein partners. The enzyme is very similar to RNP RNase P, although their substrates are different. RNase MRP’s name seems to indicate a mitochondrial location, but the latter remains controversial as the enzyme is more abundant in the nucleus than in the mitochondria [163,164]. The mitochondrial location was defined with its discovery in 1987 [163], where the authors found that it was an endoribonuclease releasing 5′-P and 3′-OH ends. The enzyme was cutting RNA near one of the transition sites of primer RNA synthesis to DNA synthesis at the leading-strand origin of mtDNA replication in vitro. They also observed that this cleavage was site-specific and that it was guided by sequences adjacent to the cleavage site that are probably conserved in mammals [163]. To reach mitochondria, all the components should be imported through the inner membrane. RMRP was proposed to cross the inner membrane of mitochondria with the help of mtPNPase [136]. However, the proteins known to interact with the RNA in the nucleus are not found, or found as traces amount in the mitochondria [165]. RMRP transit from the nucleus to the mitochondria is mediated by at least three RBPs (HuR, PNPASE and GRSF1). It has been reported that HuR bound RMRP in the nucleus, and promoted its export to the cytoplasm [166]. GRSF1 was found in mitochondrial “RNA granules” where it binds RMRP and increased its abundance in the matrix [166]. If RNase MRP is indeed imported in the matrix of mitochondria, its protein composition would be different than the one recently published, which is the RNase MRP cleaving the nuclear polycistronic ribosomal RNA [167,168]. In addition, RMRP RNA would need to be protected from mitochondrial RNases, and even if GRSF1 binds RMRP, its less than 500 amino acids length cannot protect the whole ncRNA. Convincing proof of a mitochondrial function of RNase MRP are still awaiting. Mutations in RMRP or its promoter are associated with Cartilage Hair Hypoplasia (CHH). Hallmarks of CHH are skeletal dysplasia and impaired T cell activation. Patients with mutated RMRP are susceptible to infections, a higher incidence of autoimmunity and cancer [169,170]. Robertson and colleagues (2022) linked the CHH to ribosomopathy.

### 4.3. The YbeY Ribonuclease Might Participate in the Biogenesis of Mitoribosomes

The putative endoribonuclease YbeY is widespread across eukaryotes. It is present in animals mitochondria, in kinetoplastids, plant plastids, alveolate apicoplasts and Stramenopiles (unknown localization) [171]. The enzyme has bacterial origin. Even though its highly conserved sequence and structure are well characterized, its exact function and mode of action are still the subject of much debate and investigations. A recent review was discussing YbeY, and we will briefly describe this enzyme [12].

Bioinformatics analyses determined that YbeY was part of the UPF0054 family (PFAM accession number: PF02130), and that it would therefore be a metal-dependent endoribonuclease. The enzyme possesses a conserved histidine triad with the following signature (Hx3Hx5H). In plant plastids, an additional domain is found at the C-terminus of YbeY that is a haloacid dehalogenase hydrolase-like domain [172]. The catalytic site is well conserved from bacteria: HsYbeY share 23% identity with EcYbeY and 27% of identity with *Sinorhizobium meliloti* YbeY [171]. The YbeY gene codes for a protein of 167 amino acids in humans, including the mitochondrial targeting sequence. Enzymes from different organisms are interchangeable, indicating at least a partial conserved function across phyla.

The human YbeY was proved to be catalytically active [173]. Interestingly, they also showed that HuYbeY was able to partially suppress the phenotype of *E. coli* ΔybeY mutants. Summer et al. (2020) proved the mitochondrial localization of HuYbeY and its impact on mitochondrial morphology and respiration. It seems that HuYbeY had no impact on 16S rRNA, but did have an impact on 12S rRNA, which was less expressed and less stable in HuYbeY KOs. However, no effect could be observed on the processing of RNAs in mitochondria, and rather only in vitro on different RNA substrates (including 12S rRNA fused with tRNAVal, a substrate proposed for EcYbeY) with no apparent site-specificity, suggesting that this endoribonuclease activity could be dispensable for its function in vivo [171]. By studying the HuYbeY interactome, they discovered an interaction with several ribosomal proteins including uS11m (the only one belonging to the SSU), but also p32. Further investigation of the direct and stoichiometric association of these three proteins led them to hypothesize that HuYbeY in association with p32 allowed the incorporation of uS11m into the SSU [171]. The interaction of p32 with the mitoribosome and its importance for respiration and translation has been shown previously, although the mechanisms by which this occurs remain poorly understood. The hypothesis raised here would be that it facilitated the interaction of HuYbeY (which is not negatively charged like its orthologs) with other positively charged proteins, including uS11m, to obtain SSU competent for translation initiation [12,171].

In 2021, D’Souza et al. re-emphasized the essential role played by HuYbeY in mitochondrial translation by stabilizing the SSU through incorporation of uS11m. However, unlike Summer et al., they did not find an interaction with p32 and observed a defect in mt-tRNASer (AGY) end processing in the absence of HuYbeY [174].

All in all, even if YbeY function is not completely understood, the enzyme is crucial for the organisms. Indeed, the altered expression of huYbeY or non-functional HuYbeY appears to be associated with disorders including Down syndrome, breast cancer and colorectal cancer [175,176]. In plants, the photosynthesis is impacted and the plant can probably not outcompete healthy plants in the wild [172].

### 4.4. Pet127, the Sole 5′ to 3′ Exoribonuclease Identified in Eukaryotes Mitochondria

When it comes to 5′ to 3′ exoribonucleases in mitochondria, the results are scarce in the literature compared to what is known in bacteria or in the other compartments of the cell. To date, pet127 from fungi is the sole putative 5′-3′ exoribonuclease found in mitochondria. It remained to be formally proved that it has a catalytic function [177,178,179], but a recent article showed that the pet127 PD-(D/E)XK nuclease domain is responsible for its activity in vitro. In addition, pet127 deletion and catalytic site mutants harbor the same molecular phenotype, showing that the function is linked to the catalysis of RNAs [180]. The RNA-seq study showed some intron accumulation, as well as changes in the steady-state transcriptome. In vitro results suggest that the enzyme is processive and is non-selective toward ss- or dsRNAs [180]. Lastly, the authors showed that pet127 was present in LECA and is present in different phylogenetic branches (as stated in Table 1), but lost in animals and plants.

## 5. Concluding Remarks

Mitochondrial RNases harbor different domains that can interact with their RNA substrates, as described above and summarized in Figure 2. The catalytic pockets of the enzymes are in all evidence in contact with the substrates, but additional domains in the proteins can increase or be essential for RNA substrate binding and specificity, and thus for enzyme activity. Interaction domains in trans (with protein partners) are also very important. Hence, mitochondrial RNases were often described as part of complexes with helicases or other RNA binding factors (i.e., RNA degradosomes) [21]. As reviewed here, ribonucleases act in all steps of the mitochondrial RNA lifecycle, and understanding their function and interplay, as well as their regulation, is of utmost importance to understand how the mitotranscriptome is shaped in response to environmental and cellular cues. Beyond RNases and the complexes they form with proteins partners, future research will also be necessary to determine how ribonuclease activities are coordinated with other mitochondrial gene expression processes, e.g., translation. Given the remarkable diversity of mitochondrial RNA related processes that have already been described across eukaryotes, this is expected to reveal further regulatory pathways that, in short, will all converge to control and regulate mitochondrial function, to adapt to specific environmental conditions for the respective eukaryote groups.

## Figures and Tables

**Figure 1 ijms-23-06141-f001:**
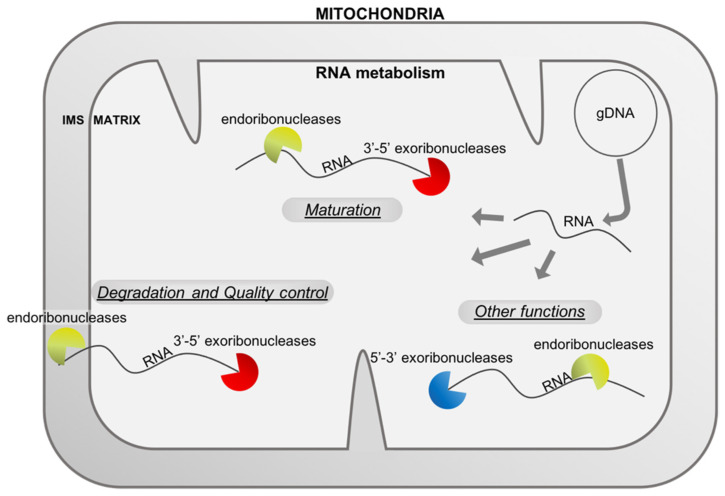
RNA metabolism in mitochondria uses different types of ribonucleases. The mitochondrial genome (here represented by a master circle at the top right) produces different types of RNAs (mRNA, rRNA, tRNA and asRNA, here stated RNA) that are matured or degraded by endo- and exo-ribonucleases. RNases are also implicated in other functions linked, for example, to the replication of the genome. All these RNases shape the final transcriptome of the mitochondria and are important for healthy mitochondria and organisms. IMS: intermembrane space, gDNA: genomic deoxyribonucleic acid, RNA: ribonucleic acid.

**Figure 2 ijms-23-06141-f002:**
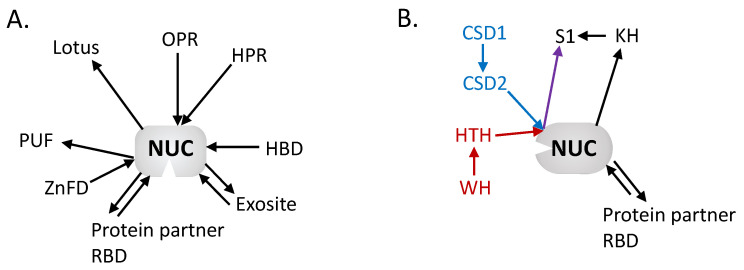
Architectural network of the mitochondria: (**A**) Endoribonuclease and (**B**) 3′-5′ Exoribonuclease domains. The network is centered on the nuclease domains with the diverse RNA binding domains surrounding. The arrows are directed from N to C termini, showing the respective position of each domain. Colored arrows have been added to discriminate different associations (red vs. blue) but that ends with same domain (purple). This highlights the diversity of organization and domains that have been recruited to perform RNA binding in mitochondrial RNases across eukaryotes. OPR: octotricopeptide repeat, HPR: heptatricopeptide repeat, HBD: hybrid binding domain, RBD: RNA binding domain, ZnFD: zinc finger domain, PUF: Pumilio/fem-3, CSD: cold shock domain, KH: K-homology, HTH: helix-turn-helix, WH: winged helix.

**Table 1 ijms-23-06141-t001:** Occurrence in the mitochondria of the different types of RNases in representative phyla of the eukaryotes. The presence (+), absence (-) of each type of enzyme, as well as the absence of information in the literature ( ), are all listed for the following phyla: Amoebozoa, Opisthokonta (Fungi and Animals), Archaeoplastida (Chlorophyta, Streptophyta, i.e., green algae and land plants), Euglenozoa (clade where trypanosomatids are found), as well as Stramenopiles (brown algae, diatoms, oomycetes, …) and Alveolata (Plasmodium, Dinoflagellates, …). The results presented in this table reflect state-of-the-art literature (reference numbers in the last column), not a de novo phylogenetic analysis of the occurrence of ribonucleases across eukaryotes. Ref. nb: reference number; RBD: RNA binding domain, RNP: Ribonucleoprotein, RNA: ribonucleic acid, PRORP: protein-only RNase P, PPR: pentatricopeptide repeat, ZS: RNase Z small, ZL: RNase Z large, ZnFD: Zinc Finger domain, PUF: Pumilio/fem-3, MRP: mitochondrial RNA processing, HBD: hybrid binding domain, RAP: RNA-binding domain abundant in Apicomplexans, HPR/OPR: hepta-/octo- tricopeptide repeat, PNPase: polynucleotide phosphorylase, KH: K-homology, WH: winged helix, HTH: helix-turn-helix, CSD: cold shock domain.

		Enzymes	RBD	Amoebozoa	Fungi	Animals	Chlorophyta	Streptophyta	Euglenozoa	Stramenopiles	Alveolata	Ref. nb
Endoribonucleases	RNase P	RNP	RNA		+	-	+/-	-	-	+	+	[7]
		PRORP	PPR	-	-	+	+	+	+	+	+	[7]
	RNase Z	ZS			-	-	-	-				[8,9,10,11]
		ZL	exosite		+	+	+	+				[8,9,10,11]
	YbeY					+			+			[12]
	MNU1/2		Lotus	-	-	-	-	spermatophta	-	-	-	[13]
	KREN		ZnFD PUF						+			[14]
	RNase MRP			+/-	+	+	+	+	+	+	+	[15,16]
	RNase H1		HBD		(+)	+		+	+			[17]
	RNase T2					+						[18]
	RAP		HPR/OPR			+			+			[19]
3′-5′	PNPase		KH-S1		-	+		+	-			[20]
	RNase II/R	DSS1	WH-HTH S1 or CSD1-CSD2 S1		+				+			[21,22]
		RNR1	CSD1-CSD2 S1					+				[23]
	REXO2					+						[24]
	RNase D								+			[25]
	PDE12					+						[26]
	KREX								+			[14,27]
	PARN							+				[28,29]
	Myg1				+	+						[30]

## Data Availability

Not applicable.

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
