# Peer review of "How RNases Shape Mitochondrial Transcriptomes"

_ijms, 2022, doi:10.3390/ijms23116141_

Round 1
Reviewer 1 Report
The review, entitled “How RNases shape mitochondrial transcriptomes”, summarized the roles of RNases in mitochondrial transcriptomes, is full of informative and detailed, excellent in organization, and strong writing skills. I have few concerns below and hope the revised version will address them.
- There was 44% of plagiarism by checking ithenticate.com.
- Regarding Table 1, I strongly suggest authors add the reference number in each presence (+) and absence (-), which guides readers to look for the original article.
- Regarding Figure 1, it’s better if author provide full name of each abbreviated.
Author Response
First, we would like to thank reviewer 1 for his time spent on the review and for his comments.
To reply to the first point, we and the editorial office of IJMS have used different softwares than "ithenticate.com" and we are not reaching the level of plagiarism that reviewer 1 is citing (44%). The level that reviewer 1 found let me think that the references were not removed from the manuscript before feeding to "ithenticate.com". In any case, few half sentences that were detected by our software, as they were similar to the original articles cited, were modified (page 10, page 14).
For the second point, in Table 1, a new column was created to add numbers corresponding to the references associated with the presence/absence of the ribonucleases in the specific phyla. Additional references were added (177-180).
For the third point, full names of acronyms were added in the legend of figure 1 that is figure 2 now (to comply to Reviewer 2 demand).
Additional modifications were done:
- A graphical abstract type figure (Figure 1)
- PET127 paragraph references were missing from the reference list and were added.
- Small errors across the manuscript were corrected
- References were added to the introduction
We think that these modifications improve the review manuscript and hope that they will satisfy reviewer 1. Thank you again for your time to verify this review manuscript.
Jérémy, Léna and Anthony
Reviewer 2 Report
This review is very interesting, it is very well written. Only minor revision are needed:
- Please enlarge the characters of table number 1, are too small
- It is recommended to create a figure that illustrates the focal points of the review
Author Response
First, we would like to thank reviewer 2 for his time spent on the review and for his comments.
To comply to reviewer 2 request we increased the size of the police but it might still remain too small as increasing the police size increase the width of the Table and this width seems to be fixed in the manuscript causing by consequence the characters to remain small. In addition, Table 1 was modified to add references linked to the presented data as per reviewer 1 request. Table 1 is therefore larger. Could the editorial office turn this Table the other way around to increase its maximal width and therefore the size of the police?
A new figure called Figure 1 was created to illustrate the focal points of the review.
Additional modifications were done:
PET127 paragraph references were missing from the reference list and were added.
Small errors across the manuscript were corrected
References were added to the introduction
References were added to Table 1
Full naming of the acronyms were added to the legend of Figure 2 (previous figure 1).
We think that these modifications improve the review manuscript and hope that they will satisfy reviewer 2. Thank you again for your time to verify this review manuscript.
Jérémy, Léna and Anthony